# Assessing the impact of group antenatal care on gestational length in Rwanda: A cluster-randomized trial

Felix Sayinzoga[1], Tiffany Lundeen[2]*, Sabine F. Musange[3], Elizabeth Butrick[2], David Nzeyimana[3], Nathalie Murindahabi[3], Hana Azman-Firdaus[2], Nancy L. Sloan[2], Alejandra Benitez[4], Beth Phillips[2], Rakesh Ghosh[2], Dilys Walker[2,5]

1 Maternal, Child, and Community Health Division, Rwanda Biomedical Center, Kigali, Rwanda, 2 Institute for Global Health Sciences, University of California San Francisco, San Francisco, California, United States of America, 3 School of Public Health, University of Rwanda, Kigali, Rwanda, 4 Department of Biostatistics, University of California Berkeley, Berkeley, California, United States of America, 5 Department of Obstetrics, Gynecology, and Reproductive Sciences, University of California San Francisco, San Francisco, California, United States of America

* tiffany.lundeen@gmail.com

**Data Availability Statement:** All data files are available from Dryad platform: https://doi.org/10.7272/Q67W69F1

## Abstract

### Background

Research on group antenatal care in low- and middle-income contexts suggests high acceptability and preliminary implementation success.

### Methods

We studied the effect of group antenatal care on gestational age at birth among women in Rwanda, hypothesizing that participation would increase mean gestational length. For this unblinded cluster randomized trial, 36 health centers were pair-matched and randomized; half continued individual antenatal care (control), half implemented group antenatal care (intervention). Women who initiated antenatal care between May 2017 and December 2018 were invited to participate, and included in analyses if they presented before 24 weeks gestation, attended at least two visits, and their birth outcome was obtained. We used a generalized estimating equations model for analysis.

### Findings

In total, 4091 women in 18 control clusters and 4752 women in 18 intervention clusters were included in the analysis. On average, women attended three total antenatal care visits. Gestational length was equivalent in the intervention and control groups (39.3 weeks (SD 1.6) and 39.3 weeks (SD 1.5)). There were no significant differences between groups in secondary outcomes except that more women in control sites attended postnatal care visits (40.1% versus 29.7%, p = 0.003) and more women in intervention sites attended at least three total antenatal care visits (80.7% versus 71.7%, p = 0.003). No harms were observed.

**Funding:** This trial is supported by the East Africa Preterm Birth initiative, a multi-year, multi-country effort generously funded by the Bill and Melinda Gates Foundation (OPP1107312). The Foundation's website is www.gatesfoundation.org. The funder reviewed the study design but did not have input on analysis or interpretation of results.

**Competing interests:** The authors have declared that no competing interests exist.

## Interpretation

Group antenatal care did not result in a difference in gestational length between groups. This may be due to the low intervention dose. We suggest studies of both the effectiveness and costs of higher doses of group antenatal care among women at higher risk of preterm birth. We observed threats to group care due to facility staff shortages; we recommend studies in which antenatal care providers are exclusively allocated to group antenatal care during visits.

## Trial registration

ClinicalTrials.gov NCT03154177

## Introduction

### Background

In 2016, the World Health Organization recommendations on antenatal care for a positive pregnancy experience prioritized research on the individual outcomes and health systems effects of group antenatal care implementation [1]. Research to date on this alternative model of antenatal care provision shows mixed results. An individually randomized controlled trial (RCT) among women in the United States at high socio-demographic risk for preterm birth showed that participation in group antenatal care was associated with a significantly lower preterm birth rate (9.8%) than participation in only individual antenatal care (13.8%) [2]. A cluster RCT among a similar cohort of women found that group antenatal care reduced the rate of small-for-gestational-age infants, and increased gestational length among small-for-gestational-age infants [3]. A meta-analysis including these studies, two other RCTs and ten observational studies found no differences in preterm birth or low birth weight [4]. However, when the authors performed a sub-group analysis by race/ethnicity limited to the two highest-quality studies, the preterm birth rate was significantly lower among African-American women who participated in group antenatal care (8.0%) compared to African-American women who participated in individual antenatal care (11.1%). Group antenatal care is hypothesized to positively impact preterm birth rates and other outcomes among women at elevated psychosocial risk due to three main features of the model: 1) greater social support between women who are linked via the group; 2) more total antenatal care-associated time spent in educational activities in facilitated group discussions; and 3) attention to key elements of person-centered care, including respect and safety, empowerment, and participation [5–7]. These elements create a more positive pregnancy care experience which may encourage antenatal care attendance and thus create additional opportunity for risk assessment by providers.

Group antenatal care has been described in several low- and middle-income country contexts. A prospective cohort trial in Ghana reported significantly higher health literacy among women who participated in group antenatal care [8]. A pilot study in Tanzania and Malawi reported feasibility, acceptability and a significant increase in attendance at five antenatal care visits, as well as significantly more satisfaction with care [9, 10]. A cluster RCT of 20 facilities in Nigeria and 20 facilities in Kenya reported that group antenatal care was associated with a significantly higher rate of facility birth in Nigeria, but not in Kenya and an increase in attendance at four antenatal care visits in both countries [11]. No study completed in a low- or middle-income country context has yet reported on the effect of group antenatal care on gestational length, preterm birth, or low birth weight.

## Objectives

The Preterm Birth Initiative-Rwanda is a partnership between investigators at the University of Rwanda and University of California, San Francisco and national health system implementors at the Rwanda Biomedical Center and Ministry of Health. Intrigued by lower rates of preterm birth among high-risk American women who participated in group antenatal care, the Preterm Birth Initiative-Rwanda aimed to test the primary hypothesis that Rwandan women receiving antenatal care at health centers that offer group antenatal care would experience increased gestational length compared to women receiving antenatal care at health centers that provide the standard, individual model of care. Stakeholders were also interested in the effect of group antenatal care on attendance, risk identification and other health outcomes.

**Context.** Rwanda's national maternity care system provided an excellent opportunity to test this innovative service delivery model due to its community capacity, cultural foundations in community-based decision-making and cooperation, and extant, longitudinal antenatal care registers. During this study period, 2017–2018, Rwanda's national guidelines prescribed that each childbearing woman be offered four focused antenatal visits (according to WHO recommendations prior to 2016) and four postnatal care visits (within 24 hours, 2–3 days,7–14 days, and 42 days of infant life). Currently, national guidelines are under revision to align with the 2016 recommendations [1].

In Rwanda, routine antenatal care is provided in government-system health centers staffed entirely by nurses and midwives. Women are required to pay a fee at the time of each antenatal visit and facility birth care. A community-based insurance scheme is available to all Rwandan families, which decreases but does not eliminate the cost of these services. Virtually all (99%) pregnant women attend at least one antenatal visit, while only 44% attend the four antenatal visits [12]. Antenatal care providers participate in a performance-based incentive program that rewards them by the proportion of pregnant women who enroll in antenatal care before 16 weeks and attend four antenatal visits following the focused antenatal schedule. While gestational age or gestational length assessments are challenging as early ultrasound is not routinely available in Rwandan health centers, 56% of women report they enter antenatal care before 16 weeks of pregnancy based on the last menstrual period [12]. 91% of births occur in a facility such as a health center or district hospital. About 19% of newborns and 43% of women receive a postnatal assessment within the first two days after birth.

This article reports the primary and secondary outcomes of this cluster RCT, including gestational length, mortality among preterm neonates, attendance at four antenatal care visits, attendance at the first antenatal care visit before 14 completed weeks gestation, attendance at a six-week postnatal care visit at a health center, identification of women as at high risk at any antenatal care visit, and caesarean section rates among enrolled women. Results of other outcomes, including women's and providers' experiences of group antenatal care, are reported elsewhere [13–15].

## Methods

### Trial design

To test the primary hypothesis that group antenatal care would extend gestational length, the Preterm Birth Initiative-Rwanda designed a cluster RCT in which a cluster was defined as a health center and the population served in its catchment area. We chose a clustered design because Rwandan stakeholders preferred to offer all women at each health center the same model of care. Clusters were pair-matched and one of each pair was assigned to continue individual antenatal visits while the other was assigned to provide group antenatal visits.

At half of the health centers included in this study we implemented community-based urine pregnancy test by community health workers and basic obstetric ultrasound by nurses and midwives to strengthen gestational age assessment and assess whether these interventions affected the secondary outcomes of attendance at four antenatal care visits and initiation of antenatal care before 14 completed weeks. Health center pairs were matched to similar pairs to ensure balance, and one pair from each quadruple was assigned the additional intervention of basic obstetric ultrasound by nurses and midwives at the health center and community-based urine pregnancy tests administered by community health workers in the catchment area.

The intervention group consisted of all health centers providing group antenatal visits; half of these health centers also provided basic obstetric ultrasound and pregnancy testing at community level. The control group consisted of all health centers providing standard individual antenatal visits, half of which also provided basic obstetric ultrasound and pregnancy testing at community level.

**Ethical approval.** Ethical approval was granted by the Rwanda National Ethics Committee (No.0034/RNE/2017) and University of California, San Francisco Institutional Review Board (16–21177). Data were collected on women aged 15 and older presenting for antenatal care at the 36 study health care centers who provided written informed consent between May 25, 2017 and December 31, 2018. This study was permitted by the Rwanda National Ethics Committee to waive parental consent for pregnant minors ages 15 and older. No pregnant adolescents younger than 15 years were documented to have been invited to participate in the study.

## Participant selection and recruitment

All pregnant women presenting for their first antenatal visit received standard individual care from a provider. Providers invited women to participate in the study and attend future antenatal visits at the study site according to the Rwanda focused antenatal care schedule. At sites randomized to group antenatal care, providers and study staff invited women to participate in group antenatal care; those who declined continued to receive individual care at the study site. After the antenatal care provider estimated the woman's due date based on the last menstrual period and symphysis-fundal height, study staff assigned the woman to a group of eight to 12 women with similar due dates (preferably within a two-week gestational-age period, with an upper limit of 4 weeks difference). Once the woman was assigned to a group, she was invited to return for three subsequent scheduled group antenatal visits at eight-week intervals, starting at 22–24 weeks gestation, and one postnatal group visit. While group antenatal services were offered to all women at group care facilities, only women who consented to study participation, presented for the first antenatal care visit before 24 completed weeks gestation, attended at least two antenatal care visits during pregnancy, and whose birth outcomes were discoverable by study staff were eligible and included in the analyses.

The primary analysis included only those women with a gestational length between 24 weeks and 43 weeks, documented by the birth care provider in the birth facility's maternity register. We included mother-infant units in the primary analysis if the infant's birth weight fell between the Intergrowth-21st Project's upper and lower centiles, by sex and gestational length [16]. The Intergrowth-21st Project provides international standards for female and male infants between the 3rd and 97th centiles by gestational length.

## Study interventions

This study included one primary (group antenatal care) and two secondary (basic obstetric ultrasound and urine pregnancy tests in the community) interventions. These interventions are described below, and more details are available in a separate publication [17].

**Group antenatal care and postnatal care.** Each group antenatal visit occurred in a room accommodating eight to 12 pregnant women, one antenatal provider and one community health worker. During the first half of these two-hour sessions, antenatal care providers met with each woman in a semi-private area of the room for brief individual assessments while the group of women socialized and participated in health assessment activities such as weighing one another and taking each other's blood pressure readings with an electronic cuff, under the supervision of the community health worker. During the second hour of the session, the provider and community health worker co-facilitated a discussion of health-related topics that aligned with the stage of pregnancy and the health issues of highest importance at that time. The full list of visit timing and topics at each visit are described elsewhere [17]. Providers and community health workers elicited concerns and questions from women and encouraged group care participants to share knowledge and support with one another.

The Rwanda group antenatal care model was customized by a Technical Working Group convened for this study, composed of representatives from maternal-child health stakeholder organizations in Rwanda [18]. During the study period, the Rwanda Ministry of Health recommended that the group care curriculum follow the four focused antenatal visits model and no more than four antenatal visits per woman could be accommodated by the health system. The Technical Working Group hoped that the social support fostered among women in the same antenatal group would continue into the postnatal period and motivate women to seek care; for this reason, a postnatal group visit was included in the model even though a postnatal visit was not expected to impact the primary outcome (gestational length). Experienced group antenatal care providers from the United States prepared six Rwandan providers with five days of training. These Rwandan providers then trained three antenatal care providers at each health center randomized to group antenatal care, with a three-day group care training session. The Rwandan trainers continued to provide the 18 group antenatal care clusters with targeted training, supervision, and mentoring throughout the study.

**Basic obstetric ultrasound by nurses and midwives at the health center.** The Preterm Birth initiative-Rwanda provided one ultrasound machine and 10 days of training for three antenatal care providers per site to the 18 clusters randomized to implement basic obstetric ultrasound. These clusters were asked to conduct a screening ultrasound examination for each woman on the day of her first antenatal visit, or soon after. The Rwanda Society of Radiologist trained these new ultrasound providers who subsequently received mentoring and supervisory visits from radiographers from the nearest district hospital.

**Urine pregnancy testing in the community.** Community health workers associated with health centers randomized to urine pregnancy tests in the community underwent an eight-hour training and were provided with urine pregnancy test kits. They were instructed to refer women with a positive pregnancy test to the health center for antenatal care services. The Rwanda Biomedical Center trained these community health workers, and each health center's community health worker supervisor supervised the community health workers in the catchment area.

## Outcomes

The primary outcome was specified with a testable hypothesis *a priori* during clinical trial registration (clinicaltrials.gov NCT03154177). The primary study hypothesis was that among women who presented for antenatal care at <24 weeks gestation and attended ≥2 antenatal visits documented at the site of study enrollment, antenatal care exposure at sites that offered group antenatal visits would increase mean gestational length by .5 weeks (with a standard deviation no larger than 4.3 weeks) compared to antenatal care exposure at sites that offered

standard antenatal visits. Gestational length was assigned by the birth care provider and recorded in the facility's maternity register. Birth care providers used all available data to make this determination at the time of birth (i.e. last menstrual period, birth weight, infant maturity), but gestational length assignment was not standardized or monitored. Birth weight was recorded by birth providers before leaving the delivery room using an analog infant scale; this measurement was abstracted from the maternity register at each facility. Birth care providers were not informed about the study's primary hypothesis nor primary outcome.

The secondary outcomes were adherence to the recommended four antenatal visits, gestational length at first antenatal visit, incidence of preterm birth, proportion of women delivering by caesarean section, proportion identified as at risk during antenatal care, and adherence to six-week postnatal visit. We intended to examine newborn morbidities but our data sources were not adequate to do so. Antenatal visits were counted in the longitudinal antenatal register at each facility. Gestational length at the time of the first antenatal visit was documented by the antenatal provider in the facility antenatal register. Preterm birth was assigned to any infant whose gestational length was recorded in the maternity register as less than 37 weeks. We also examined the effect of group antenatal care on mode of birth and the effect of basic obstetric ultrasound and community-level urine pregnancy tests on antenatal care attendance. Gestational length at the time of the first antenatal visit was documented by the antenatal care provider in the facility antenatal register. Antenatal visits were reported in the pre-existing longitudinal antenatal register at each facility. Postnatal visits were reported in the recently established postnatal register at each facility.

**Data sources and collection.**   On the day women consented to study participation, immediately after the first antenatal visit, all study participants completed an enrollment questionnaire about socio-economic and health history, which was administered by a health provider or study staff member. Data were entered directly on mobile tablets by data collectors into an encrypted, web-based system called Research Electronic Data Capture (REDCap) platform [19].

Study staff abstracted all data from standard Ministry of Health primary (paper) data sources located in the study facilities. These included antenatal, birth, neonatal, and postnatal registers, and individual longitudinal antenatal and postnatal records. Birth registers were reviewed at all 36 health center study sites and the six district hospitals to which those health centers referred women for higher-level care. Study staff were instructed to extract newborn morbidities from health center data drawn from community health worker SMS reports, but in practice these data were not available.

The study initially created and used a multisite Research Electronic Data Capture database. However, due to poor connectivity and the quantity of data collected per participant (367 variables), the process of syncing or uploading data from the tablets to the server was frequently disrupted. The disruption led to many duplications and erroneous linkage of study events. Upon discovery of this error, the study team substantially revised the database and conducted a thorough process to rectify any errors and retain the most complete record for each patient. Unrectified records were excluded from the final datasets. The electronic data were maintained on secure systems with access limited to the principal investigators, study epidemiologist, and designated study staff. The data were converted into SPSS for Windows version 25 [20] and STATA SE version 16 [21] for analyses.

## Sample size

At the design stage of this trial, we assumed an ICC of 0.01, and a possible effect size of half a week in gestational length based on a Cochrane review of conventional versus group antenatal

care [22]. We assumed a loss-to-follow-up rate of 30%, based on the assumptions that 10–15% of all pregnancies end in miscarriage and 15–20% of women might deliver at another location. We calculated that for a two-tailed test, α = 0.05, 1-β = 80%, and a balanced 1:1 ratio of intervention (group antenatal care) to control (standard antenatal care) study participants, a minimum sample of n = 1,163 eligible women per study group (intervention and control) was required. The sample size was increased to account for a cluster design effect of 3.21 (cluster randomization to study group by health center rather than by individual) and to account for a loss to follow-up rate of up to 50%, for a total required sample size per study group of 3,640, or an average of 202 women per health center at 36 health centers [23]. As descriptive outcomes, the secondary outcomes were assessed in the available sample without hypothesis testing or multiple comparisons adjustment.

### Site selection

Five of 30 districts in Rwanda, including Burera, Bugesera, Nyamasheke, Rubavu, and Nyarugenge, were selected by the Ministry of Health as locations for this cluster RCT based on their service capacity and need. To gather information that would inform study site selection, data collectors visited all 55 public health centers in the study districts and interviewed staff about the facility itself, human and material resources, and client volumes. We selected 36 health centers for inclusion that 1) had historical volumes of women that made organizing groups feasible (35–125 births at the health center per month); 2) reported at least two antenatal care providers per day when antenatal care was offered, to increase the likelihood that one antenatal care provider could be exclusively allocated to scheduled group visits; and 3) had a room sufficiently large for group visits.

## Randomization

To optimize study group comparability, 36 selected health centers were pair-matched. First, the study statistician applied a non-bipartite matching algorithm in R, using the nbpMatching package (available from: https://cran.r-project.org/web/packages/nbpMatching/index.html), including health center-specific data on the monthly number of women registering for antenatal care, monthly number of births, proportion of first antenatal visits completed before 16 weeks gestation, and availability of screening tests as a composite. Thirty sites were matched with the non-bipartite matching algorithm [24] with the remaining six presented as multiple options, which were then reviewed by the study team that finalized matching based on the monthly number of women registering for antenatal care, rural/urban designation and distance to the nearest hospital. After matching was complete, pairs were further matched to quadruples. Then one site in each pair was assigned to group antenatal care and the other to standard antenatal care using random selection in R software [25]. One pair of each quadruple was similarly randomly selected for implementation of basic obstetric ultrasound and urine pregnancy testing in the community.

No allocation concealment was used. The Ministry of Health notified the heads of the health centers of their selection status; all selected health centers agreed to participate.

### Statistical analyses

Cleaning (range and logic) checks were applied, and eligibility and critical outcomes data requiring further cleaning were sent to designated field staff to review and resolve discrepancies. We conducted individual level bivariate analyses stratified by study group to assess study group comparability.

We compared the control group (individual antenatal visits) with the intervention group (group antenatal visits) in the primary analysis. The primary analysis was conducted using gestational length as recorded by the maternity providers after conducting sensitivity analyses to determine if any other gestational length variable or computation would produce results that were more plausible for correct gestational length classification. We relied on the Intergrowth 21[st] Project's international standards for birth weight by sex and gestational length to determine the plausibility of correct GA classification, excluding infants with birth weights outside the upper and lower centiles.

Treatment effect was estimated using linear and logistic regression generalized estimating equations with robust variance estimation to account for clustering of births within facility and to adjust for pairing of facilities. We used exchangeable correlation structure and conducted an individual level analysis. We developed a Directed Acyclic Graph to identify potential confounders and mediators for which data were available. In descriptive analyses, we investigated the factors on which the intervention and control arms were statistically significantly different. The final model was adjusted for pairing and clustering only. In sensitivity analyses, we additionally adjusted for potential confounders or mediators to demonstrate that the results were unaffected by residual confounding that may have resulted due to unbalanced study arms. Analogous analyses were conducted to assess continuous and categorical secondary outcomes. Significance tests were two-tailed at the 5% level. Analyses were conducted using SPSS v25 [17, 20] and STATA v16 [21].

## Results

The Consolidated Standards of Reporting Trials (CONSORT) diagram is presented as Fig 1 showing that 25,258 women consented to participate in the study. Of those, 9,420 were excluded because they presented after 24 weeks or did not attend two antenatal visits at the facility or for other reasons. The paired randomization by health center produced a nearly equal number of women presenting for antenatal care across the study groups. The average number of women recruited per health center was similar between study groups (mean 695, range 146–1015 in control clusters and mean 708, range 194–1090 in intervention clusters). In total, 3,918 women were lost to follow-up (i.e., no birth outcome captured), with similar numbers lost in each study group. Prior to analysis, 3,077 women were eliminated because of missing or implausible outcome data. Thus, 8,843 women were included in the primary analysis because they had gestational length outcomes between 24–43 weeks documented by March 31, 2019 and the infants' birth weights were plausible using the upper and lower centiles of the Intergrowth 21[st] Project's standards. There were 4,091 eligible mother-baby units in the control group and 4,752 eligible mother-baby units in the intervention group.

### Facility characteristics

Based on the design and data obtained for matching facilities before the intervention began, facilities randomized to the control and intervention conditions were similar (Table 1).

### Participant characteristics

While various differences between the study groups' participant characteristics were statistically significant, we do not consider these differences to be practically or clinically significant (Table 2). Fewer women in the intervention condition had health insurance, but among women in the lowest income category, more were enrolled in the community-based insurance system. Women in the intervention condition had higher levels of education, more were nulliparous, and more were short of stature (height <150 cm) or had a small middle-upper arm

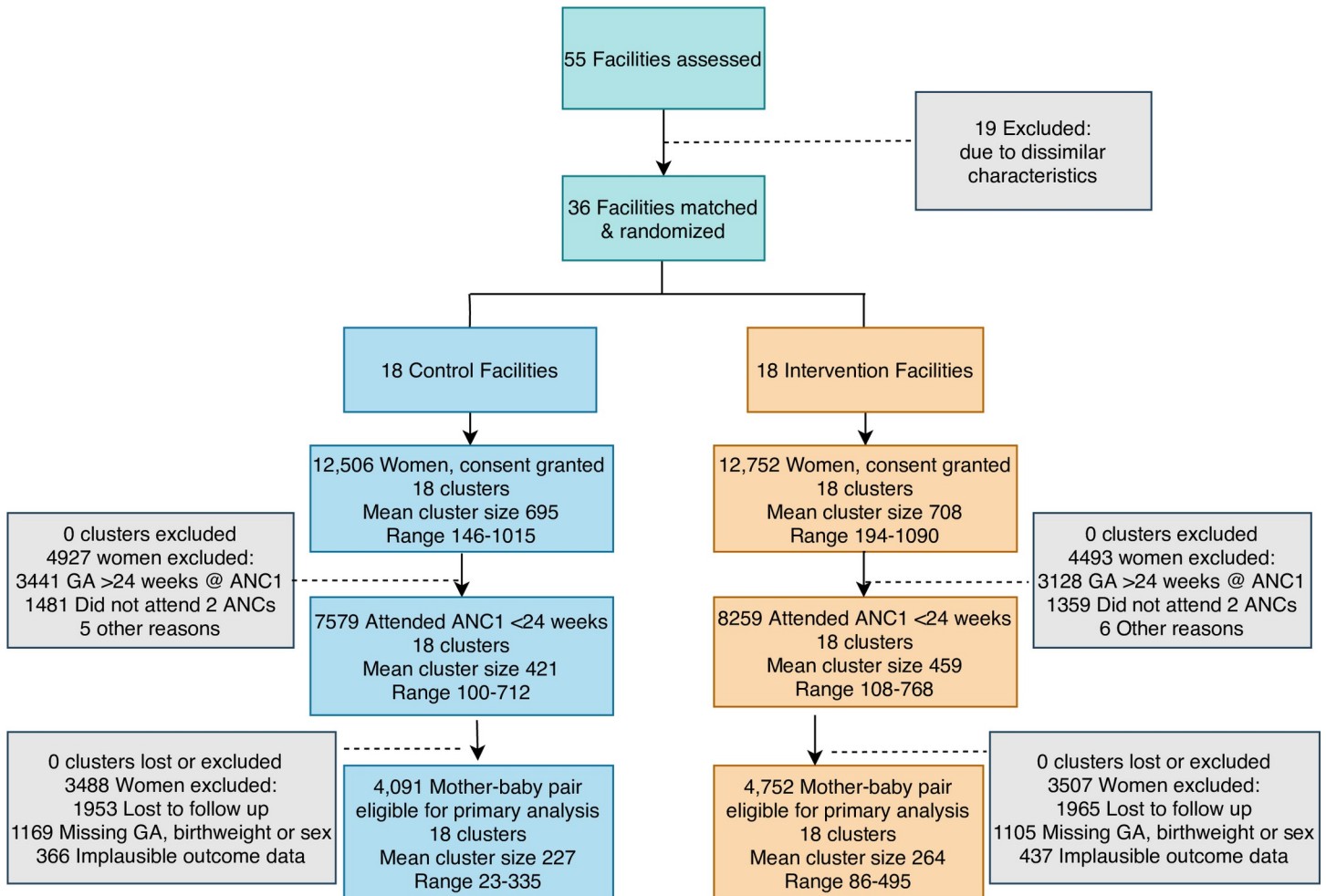

**Fig 1. Consolidated Standards of Reporting Trials (CONSORT) diagram for the Preterm Birth Initiative-Rwanda trial of group antenatal care.**

circumference (<21cm) when compared to women in the control condition. Fewer women in the intervention condition were over 35 years of age or smoked tobacco, while more women in the intervention condition were HIV-positive. Among multiparous women, more in the intervention condition reported a history of preterm birth or stillbirth. These differences were incorporated into the adjusted analyses of the primary and secondary outcomes.

**Table 1. Baseline facility characteristics by study group.**

| | Control (n = 18) | Intervention (n = 18) |
|---|---|---|
| | Mean (range) / % | Mean (range) / % |
| Births per month at the facility | 57 (32–123) | 50 (29–99) |
| Women per month who enroll for antenatal care at the facility | 98 (54–183) | 97 (58–248) |
| Mean proportion of pregnant women who attended the first antenatal care at gestational age less than 16 completed weeks (range) | 45% (21%-96%) | 53% (25%-88%) |
| Basic antenatal screening tests available, out of 5 | 4.7 (3–5) | 4.5 (3–5) |
| Proportion of facilities considered to be in a rural setting | 83% | 72% |
| Distance to referral hospital, in kilometers | 31 (10–60) | 32 (5–80) |

**Table 2. Women's characteristics at the first antenatal visit, non-missing observations.**

| Characteristics of Women | Control | | Intervention | |
|---|---|---|---|---|
| Age | N | % | N | % |
| <18 | 77 | 1.0 | 75 | 0.9 |
| 18–35 | 6165 | 81.9 | 6961 | 85.6 |
| >35 | 1283 | 17 | 1070 | 13.2 |
| **Household socio-economic status: "Ubudehe" category**[*] | | | | |
| Category 1 (poorest) | 892 | 18·3 | 943 | 19.1 |
| Category 2 | 1986 | 40·7 | 1904 | 38.6 |
| Category 3 | 1754 | 35.9 | 1620 | 32.9 |
| Category 4 (richest) | 3 | 0.1 | 6 | 0.1 |
| I don't know | 48 | 1.0 | 160 | 3.2 |
| None | 198 | 4.0 | 297 | 6·0 |
| **Currently has health insurance** | | | | |
| No | 488 | 9.4 | 979 | 19.8 |
| Yes | 4692 | 90.6 | 3972 | 80.2 |
| **Education level** | | | | |
| None | 691 | 9.2 | 498 | 6.1 |
| Some primary | 3100 | 41.2 | 2900 | 35.3 |
| Completed primary | 1957 | 26.0 | 2398 | 29.2 |
| Some secondary | 995 | 13.2 | 1290 | 15.7 |
| Completed secondary | 625 | 8.3 | 837 | 10.2 |
| Some college or university | 65 | 0.9 | 128 | 1.6 |
| Completed college or university | 95 | 1.3 | 158 | 1.9 |
| **Work outside her home** | | | | |
| Unemployed | 650 | 8.6 | 938 | 11.4 |
| Professional/technical/managerial | 115 | 1.5 | 147 | 1.8 |
| Clerical | 5 | 0.1 | 4 | 0.0 |
| Sales and services | 433 | 5.7 | 439 | 5.3 |
| Skilled manual | 14 | 0.2 | 32 | 0.4 |
| Unskilled manual | 187 | 2.5 | 248 | 3.0 |
| Domestic service | 143 | 1.9 | 609 | 7.4 |
| Agriculture | 5923 | 78.2 | 5683 | 68.8 |
| Other | 61 | 0.8 | 100 | 1.2 |
| Missing | 48 | 0.6 | 59 | 0.7 |
| **Cooking fuel at home** | | | | |
| Wood & Charcoal | 7443 | 98.6 | 8039 | 97.9 |
| Electricity, Kerosene, Gas | 104 | 1.4 | 176 | 2.1 |
| **Risk factors for preterm birth** | | | | |
| Height < 150 cm | 239 | 3.2 | 375 | 4.6 |
| Middle-upper arm circumference < 21 cm | 164 | 2.2 | 354 | 4.3 |
| Previously known human immunodeficiency virus status positive | 102 | 1.7 | 136 | 1.9 |
| Presence of a household smoker | 372 | 4.9 | 326 | 4.0 |
| Nulliparous | 1873 | 25.5 | 2472 | 30.2 |
| History of diabetes | 21 | 0.3 | 28 | 0.3 |
| History of hypertension | 37 | 0.5 | 35 | 0.4 |
| History of stillbirth | 88 | 1.6 | 137 | 2.4 |
| History of infant with low birth weight | 184 | 2.4 | 132 | 1.6 |
| History of preterm birth | 16 | 0.3 | 38 | 0.7 |

(*Continued*)

**Table 2.** (Continued)

| Characteristics of Women | Control | | Intervention | |
|---|---|---|---|---|
| Elevated blood pressure today | 6 | 0.1 | 15 | 0.2 |
| Anemia | 14 | 0.2 | 23 | 0.3 |
| Multiple gestation diagnosed today | 192 | 2.5 | 152 | 1.8 |
| None reported | 6985 | 92.2 | 7651 | 92.6 |

*An Ubedehe category is assigned to each household. Local community members at the cell level are required to gather community members together and, with the help of Ubudehe facilitators/trainers, identify and place community members into different economic categories, ranging from the poorest households (lowest category) to the richest households (highest category) [26].

## Outcomes

**Primary outcome.** The primary outcome, gestational length, was identical in the intervention and control groups (39.3 weeks (SE 0.2) and 39.3 weeks (SE 0.2), respectively; Table 3). The intra-cluster correlation coefficient was 0.00063. When we examined a subset of women in both study conditions (group antenatal care versus standard antenatal care) who also received an ultrasound examination between 6 and 22 weeks gestation and we calculated gestational length using that ultrasound data, the mean gestational length remained similar between groups (39.8 weeks (SE 0.03) versus 39.5 weeks (SE 0.03), respectively; S1 Table).

**Secondary outcomes.** Secondary outcomes among women in control and intervention groups are included in Table 3. There were no significant differences between study groups in

**Table 3. Distribution of the primary and secondary outcomes by study groups and the effect of the intervention on these outcomes.**

| Primary Outcome | Control | | Intervention | | | |
|---|---|---|---|---|---|---|
| | $n_1/n_2$ | Mean (SE) | $n_1/n_2$ | Mean (SE) | $\beta$(95% Confidence Interval)[a] | P-value |
| **Gestational length (in weeks)** | **4091** | **39.3 (0.02)** | **4752** | **39.3 (0.02)** | **-0.07 (-0·18, 0·04)** | **0.24** |
| Secondary outcomes | $n_1/n_2$ | % | $n_1/n_2$ | % | Odds ratio (95% Confidence Interval) [a] | P-value |
| Preterm birth | 146/4091 | 3.6 | 177/4752 | 3.7 | 1.06 (0·73, 1.53) | 0.77 |
| Cesarean birth | 444/4084 | 10.9 | 573/4745 | 12.1 | 1.00 (0·78, 1.28) | 0.98 |
| Proportion of women who attended at least 4 antenatal visits | 2654/7579 | 35.0 | 3478/8259 | 42.1 | 1.20 (0·86, 1.68) | 0.29 |
| Proportion of these women who attended 1st antenatal visit before 14 completed weeks gestation | 3148/7516 | 41.9 | 3040/8223 | 37.0 | 0.86(0.61, 1.21) | 0.39 |
| Mortality among preterm neonates (among all preterm babies) | 14/146 | 9.6 | 13/177 | 7.3 | [b] | |
| Attendance at postnatal visits at 6 weeks (among all eligible women with delivery outcome) | 1675/4091 | 40.1 | 1413/4752 | 29.7 | 0.52 (0.33, 0.80) | 0.003 |
| Women identified at high risk (among all eligible women, including any high risk in pregnancy)[c] | 623/4091 | 15.2 | 729/4752 | 15.3 | 0.98 (0.71, 1.35) | 0.89 |

[a] Models are adjusted for pairing and clustering of births within facilities.

[b] Sufficient observations are not available for the models to fit.

[c] Any high risk in pregnancy includes history of low birth weight, preterm birth, stillbirth, neonatal death, or detection of anemia, edema, proteinuria, not gaining weight, high bp, multiples, abnormal lie after 32 weeks or other provider initiated referral to higher level care.

$n_1$ = Numerator for the specific category in the control/intervention arm

$n_2$ = Total number of non-missing observations for the respective variable in the control/intervention arm.

preterm birth or mode of birth by Cesarean. There was no difference between groups in attendance at four antenatal visits and first antenatal visit before 14 completed weeks of pregnancy. Mortality among preterm neonates was slightly lower in the intervention group, however the number of data points in this category was insufficient to perform inferential statistics. Attendance at the six-week postnatal visit was higher in the control group than the intervention group. There was no difference in the proportion of women identified as having a risk factor during pregnancy between the two study groups.

Exploratory analysis revealed no difference in low birth weight between groups (S3 Table). Further adjusted analyses for factors associated with preterm birth and low birth weight did not change the results, except that multiple gestation and maternal weight less than 45 kilograms or over 80 kilograms were associated with higher low birth weight in the intervention group.

Among women with subjective or measurable risk factors for poor perinatal outcomes ("high-risk women"), the odds ratio for low birth weight was significantly higher among women who received antenatal care at clusters randomized to group antenatal care compared to those at clusters randomized to standard antenatal care, but only among women with maternal pre-pregnant weight less than 45 kilograms or more than 80 kilograms or with multiple pregnancy (S2 Table). There were no other differences in outcomes for "high-risk women" between study conditions with respect to gestational length, preterm birth, or mode of delivery.

Exploratory analysis to better understand attendance patterns showed that more women who attended antenatal care at facilities randomized to group care attended at least three total antenatal visits compared to women in the control group (80.7% versus 71.7%, p = 0.003) and that the mean number of antenatal visits attended was slightly higher (3.19 versus 3.05, p = 0.05; S3 Table).

There was no difference in gestational age at entry to antenatal care or attendance at four antenatal visits among women who attended antenatal care at sites that implemented urine pregnancy tests in the community and basic obstetric ultrasound at the health center compared to those without these interventions (Table 4).

## Discussion

In this cohort of 8,843 women who attended an average of three antenatal visits, we did not find a difference in mean gestational length between study groups. Fewer than 4% of births were classified as preterm, but this result is substantially lower than other estimates for Rwanda [27] and is likely influenced by multiple biases inherent in gestational length misclassification. We used gestational length at birth as documented by the maternity provider in the facility register as our endpoint. This variable, when compared to all the gestational length-related data items available in our data set, resulted in the highest proportion of infants whose

**Table 4. Antenatal care attendance among eligible women who presented at <24 weeks and attended at least two antenatal visits, comparing women from clusters randomized to Urine Pregnancy Testing (UPT) and ultrasound (US) to women from clusters randomized to no UPT by community health workers nor US at the health center level.**

|  | Yes UPT and US | | No UPT nor US | | Odds ratio (95% Confidence Interval) * | p-value* |
|---|---|---|---|---|---|---|
|  | $n_1/n_2$ | % | $n_1/n_2$ | % | | |
| Proportion of women who attended at least 4 antenatal visits | 2876/7104 | 40.5 | 3256/8734 | 37.3 | 1.07(0.85 1.35) | 0.57 |
| Proportion of these women who attended 1st antenatal visit before 14 completed weeks gestation | 2899/7037 | 41.2 | 3289/8702 | 37.8 | 1.07 (0.86, 1.34) | 0.54 |

birthweights were plausible by Intergrowth 21$^{st}$ Project's standards by infant sex. However, by applying these birth weight standards we excluded 26% of infants that had been classified as preterm because their birth weights were deemed implausible. This exclusion also resulted in fewer than 3% of infants with low birth weight, which is also inconsistent with reports from Rwanda [12]. These misclassification issues may have increased the similarity of gestational length at birth across groups.

In addition, the gestational length variable was 28% more likely to be missing for infants not born in the health center at which the mother attended antenatal visits; infants born outside the health center were born at the referral hospital, private facilities, or at home and may have disproportionately included preterm infants. Women with pregnancy risk factors identified by providers at the first antenatal visit were routinely referred to the district hospital for surveillance, and these high-risk women were less likely to continue antenatal care at the health center; this may have skewed our sample toward a more low-risk population with lower incidence of preterm birth and low birth weight.

To our knowledge, this is the largest-scale implementation of group antenatal care in an low- middle-income country context to date and we relied on existing register data documenting maternal and infant outcomes. This intervention did not include any additional provider staff, patient incentives or extra communications to women enrolled in the study.

The intervention dose was limited to about two group antenatal visits (three total antenatal visits), on average, among women in the intervention clusters. We hoped eligible women would attend three group antenatal visits during pregnancy according to the focused four-visit antenatal care model advocated by WHO before 2016, but it appears that the barriers to antenatal care attendance were greater than the potential appeal of the group care model. We hypothesize that two main barriers influenced women's antenatal care attendance. First, women who presented for antenatal care at health centers were expected to enroll their families in the community-based insurance program and pay an annual premium (according to their income level) and co-pay at each visit according to their income level. If women do not enroll in community-based health insurance, they are obliged to cover the total cost of all facility services. Women and their families are required to present vital records and proof of income (Ubudehe category) to enroll in the insurance program. Before each antenatal visit, women are required to make a co-payment, which many of them find difficult to pay and thus could be a barrier to antenatal care attendance [14]. The second barrier to antenatal care attendance may paradoxically be related to the performance-based incentive program in place at these health centers. Antenatal care providers are financially incentivized when women in the catchment area present for the first antenatal visit before 16 weeks and then attend the following three antenatal visits within the gestational age ranges recommended by the focused antenatal care model. Visits outside the narrow gestational age ranges suggested in the focused four-visit antenatal care model were either not allowed or when they actually occurred they were not documented.

As reported in Table 3, we did not find a difference in gestational length among Rwandan women who attended an average of three total antenatal visits (one standard visit and two group visits in the intervention condition) compared to an average of three total standard antenatal visits (control condition). The intervention dose may have been too low to have an effect on health outcomes. A 2016 cluster RCT of group antenatal care in the United States reported that attendance at five or more group visits (of ten group visits offered in the intervention condition) was associated with improvements in all measured outcomes [3]. The minimum therapeutic dose is likely different in each context, but it appears that a dose of at least five group antenatal visits should be studied when health outcome endpoints are of interest. Grenier et al reported that Kenyan and Nigerian women who received care at facilities randomized to

group antenatal care attended a total of six antenatal visits (median) [11]. We propose that future research of group antenatal care in low- and middle-income country contexts should study the effects of at least five group antenatal visits, within the WHO 2016 antenatal care framework of at least eight total antenatal contacts offered to pregnant women.

While there was a statistically significant difference regarding attendance at the six-week postnatal visit, it favored the control rather than the intervention group. There may be several potential reasons for this. First, a six-week postnatal care visit at facilities was not as clearly established in Rwanda as antenatal care patterns were. Thus, we are unclear whether postnatal measurements in all facilities were practically measuring the same thing, or whether women in control visits may have had immunization visits for their infants counted as postnatal care visits whilst group care sites only counted group visits. Second, women themselves may have valued a brief immunization focused visit rather a longer group session. Therefore, as group postnatal visits were set for a specific date from the time of the first antenatal visit, it is possible that a broader distribution of actual delivery dates than initially expected made postnatal care visit timing inconvenient or inappropriate for some women. We recommend further research in this area for others implementing group postnatal visits.

The change to group antenatal care provision at the 18 intervention health centers was disruptive, as any change will be. The most common concerns expressed by group antenatal care providers were: 1) the staffing model at these health centers often required that antenatal care providers also cover the maternity service, where women arrived in labor and needed immediate attention even if a group antenatal visit was in session; 2) both women and providers found it challenging to start the group antenatal visits at the scheduled time; and 3) the group care model required additional staff time and attention to create a group visit plan that would work, month to month, within the operations of the health center [13]. Without additional human resources to manage these challenges, antenatal care providers had to balance their enthusiasm for the group antenatal care model with the burden of additional tasks. We report in a separate publication that the group antenatal care model was delivered at approximately 80% fidelity across more than 2,600 documented visits [15]. The lowest-rated fidelity outcome was keeping to the intended time frame for each group visit, which was related to antenatal care staff being called to other services, especially labor and birth, when facilities were short-staffed.

## Limitations

The variable we used for the primary outcome is documented by maternity providers in whole weeks, which limited the granularity of the analysis for differences between study groups. The generalizability of these results may be limited by the fact that our sample was restricted to women who registered for antenatal care before 24 completed weeks and attended at least two antenatal visits, which may represent a low-risk population of women in this context. We do not have outcome data for 24.7% of our cohort, which may also limit the generalizability of these findings; data transmission errors related to network connection problems contributed to this loss. We encountered substantial difficulties using a web-based data collection system in remote areas that had unstable connectivity. Finally, while use of facility data overall could be seen as a strength, we found that data collector access to newborn morbidity was not a reliable way to collect this information and were unable to complete analysis of this secondary outcome.

## Conclusion

The group antenatal care model implemented in our study did not result in a difference in gestational length or preterm birth rate in the intervention group compared to the control group.

In order to understand whether this intervention will improve health outcomes for other populations of women, we suggest follow-up studies of both the effectiveness and costs of higher doses of group antenatal care among women at a higher baseline risk of preterm birth and only in facilities in which antenatal care providers are exclusively and reliably allocated to group antenatal care provision during scheduled group visits.

## Supporting information

**S1 Table. Gestational length, incidence of preterm birth, and incidence of low birth weight among a subset of women; gestational length calculated by ultrasound-adjusted gestational age when ultrasound examination was completed between 6 and 22 weeks gestation.**
(DOCX)

**S2 Table. Adjusted analysis for maternal characteristics associated with selected outcomes, using the control group as the reference.**
(DOCX)

**S3 Table. Distribution of secondary outcomes by study groups and the effect of the intervention on these outcomes.**
(DOCX)

**S1 File.**
(DOCX)

**S2 File.**
(DOCX)

## Acknowledgments

First and foremost, we would like to acknowledge the women and their infants who provided consent to participate in this study. We would also like to thank the providers and community health workers who participated in this study, whose daily work is humbling to us all. We are grateful to the Rwanda Ministry of Health and Rwanda Biomedical Center and the head of health center at each of the 36 study facilities. We are grateful for the many people who contributed to this project over time, including Janine Condo, Jean-Baptiste Byiringiro, Catherine Mugeni, Evodia Dushimimana, Andrew Muhire, Vedaste Ndahindwa, Lauriane Nyiraneza, Nicole Santos, Caroline Kusi, Yvonne Delphine Nsaba Uwera, Antoinette Kambogo, Angele Musabyimana, Alice Umukunzi, Olive Tengera, Athanasie Mbuguje, Grace Liu, Matthew Meeks, Hannah Park, Mona Sterling, Fidens Dusabeyezu, and Wenjing Zheng. We thank our program officers France Donnay, Janna Patterson, Jerker Liljestrand, and Manpreet Singh for their guidance, encouragement, and support. We are grateful for the members of the Preterm Birth Initiative East Africa External Advisory Committee, our colleagues at the California Preterm Birth Initiative, and the members of our joint Strategic Advisory Board.

## Author Contributions

**Conceptualization:** Felix Sayinzoga, Sabine F. Musange, Elizabeth Butrick, Dilys Walker.

**Data curation:** Elizabeth Butrick, Nathalie Murindahabi, Hana Azman-Firdaus, Beth Phillips.

**Formal analysis:** Elizabeth Butrick, Nancy L. Sloan, Alejandra Benitez, Beth Phillips, Rakesh Ghosh.

**Investigation:** Felix Sayinzoga, Sabine F. Musange, Elizabeth Butrick, Dilys Walker.

**Methodology:** Felix Sayinzoga, Tiffany Lundeen, Sabine F. Musange, Elizabeth Butrick, David Nzeyimana, Nancy L. Sloan, Alejandra Benitez, Dilys Walker.

**Project administration:** Felix Sayinzoga, Tiffany Lundeen, Sabine F. Musange, Elizabeth Butrick, David Nzeyimana, Nathalie Murindahabi, Hana Azman-Firdaus, Beth Phillips.

**Supervision:** Felix Sayinzoga, Sabine F. Musange, Dilys Walker.

**Visualization:** Elizabeth Butrick.

**Writing – original draft:** Tiffany Lundeen, Elizabeth Butrick, Nancy L. Sloan.

**Writing – review & editing:** Felix Sayinzoga, Sabine F. Musange, David Nzeyimana, Nathalie Murindahabi, Hana Azman-Firdaus, Alejandra Benitez, Beth Phillips, Rakesh Ghosh, Dilys Walker.

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
