## [Decision Letter · Decision Letter 0]

14 Jul 2020

PONE-D-20-10387

Assessing the impact of group antenatal care on gestational length in Rwanda: a cluster-randomized trial

PLOS ONE

Dear Dr. Lundeen,

Thank you for submitting your manuscript to PLOS ONE. After careful consideration, we feel that it has merit but does not fully meet PLOS ONE’s publication criteria as it currently stands. Therefore, we invite you to submit a revised version of the manuscript that addresses the points raised during the review process.

We look forward to receiving your revised manuscript.

Kind regards,

Seth Adu-Afarwuah

Academic Editor

PLOS ONE

Journal Requirements:

2. You indicated that you had ethical approval for your study.

In your Methods section, please ensure you have also stated whether you obtained consent from parents or guardians of the minors included in the study or whether the research ethics committee or IRB specifically waived the need for their consent, or whether minors with children are legally allowed to consent for themselves.

4. Please note that in order to use the direct billing option the corresponding author must be affiliated with the chosen institute. Please either amend your manuscript or remove this option (via Edit Submission).

5. We noted in your submission details that a portion of your manuscript may have been presented or published elsewhere:

'Table 1 was previously published in the Journal of Midwifery and Women's Health. Figure 1 was previously published in Gates Open Research. These publications are included as Related Manuscripts in this submission.'

Please clarify whether this publication was peer-reviewed and formally published.

If this work was previously peer-reviewed and published, in the cover letter please provide the reason that this work does not constitute dual publication and should be included in the current manuscript.

Reviewers' comments:

Reviewer's Responses to Questions

**Comments to the Author**

1. Is the manuscript technically sound, and do the data support the conclusions?

Reviewer #1: Partly

Reviewer #2: Partly

Reviewer #3: Yes

2. Has the statistical analysis been performed appropriately and rigorously? 

Reviewer #1: I Don't Know

Reviewer #2: No

Reviewer #3: Yes

3. Have the authors made all data underlying the findings in their manuscript fully available?

Reviewer #1: Yes

Reviewer #2: Yes

Reviewer #3: No

4. Is the manuscript presented in an intelligible fashion and written in standard English?

Reviewer #1: Yes

Reviewer #2: Yes

Reviewer #3: No

5. Review Comments to the Author

Reviewer #1: Thank you for the opportunity to read a well written paper on an important topic: addressing ANC and maternal/infant health in LMIC's/economically disadvantaged populations is very important. I do have some experience in healthcare in LMIC's in the African continent (but not Rwanda), cluster RCT's and GEE. I appreciate the complexity of the design, and the detailed explanations. I will focus more on technical issues related to design and analysis as this topic is not my clinical area of expertise.

1. This is a complicated design. I can not recall seeing an example of a cluster RCT with a factorial component. I remain unclear on aspects of your analysis. Usually a factorial design will report covariates for each factor, and an interaction estimate (between factors). The interaction effect is hopefully null, and then each factor effect can be reported independently. Was this done? Within a GEE, it is beyond my technical grasp how this would be setup given additional covariate for estimating random effects (intraclass correlations) as described in the manuscript. I think additional explanation is required, and perhaps supplemental analytical detail.

2. The pair matching is an additional complicated aspect of the design. As I read your methods, it was unclear to me if the 4 factors for matching were considered separately, or were considered as a composite. I think the 'range' of values which were considered a 'match' should be reported (e.g. 50%+/-5% was considered a match for proportion of first ANC visits, monthly delivery volume was 60 +/-10, et cetera). If matching variables were treated independently (and not as a composite) I'm not certain of the validity of reporting variables used in matching in Table 2 (unless to confirm no failure in matching) as variables used for matching must be within the specified parameters of the original matching unless there was a technical failure. I believe this would apply to the first 4 variables in Table 2.

3. There is a risk of selection bias given the loss to follow-up. It is reassuring that the relative proportions reported in Figure 2 are somewhat similar. Was there any analysis performed on characteristics between mothers who completed the study compared to those where mother-baby pair data was available?

4. Why was GL/GA not calculated based on the estimate from the first ANC rather than based at the time of delivery? Given entry into the study based on this estimate, and issues of blinding (although assessors were unblinded, it is an objective measure of GL/GA estimate if using the estimate from first ANC and date of delivery).

Minor comment

1. In the results section, I think more technically accurate language would be that participant characteristics were statistically significant, but were not considered to be practically or clinically significant (I'm not sure of a technical meaning for insubstantial). I'm not a fan of statistical testing of demographic features between treatment arms, and your study might be a good example of permitting the reader to draw their own conclusion as to if any of these features are practically significant (rather than being drawn to a p value that perhaps better reflects size of the study).

Reviewer #2: Abstract:

- since this was a cluster randomised trial, the reported findings should also have the number of clusters included in the analysis. You should report this alongside the numbers of women in each arm, e.g. "4091 women in X clusters and 4752 women in Y clusters at control and intervention facilities, respectively, were included..."

- you need to specify the statistics represented by the "±1·6" and "±1·5" - is this the range of values? If so say so; if it is another statistic other than the range, please remove the ± designation.

Objectives:

- the objectives make no mention of the ultrasound component of the intervention; indeed having read it, the description of the trial seems incomplete to me - it is a factorial cluster randomised trial testing the effect of group ANC with or without obstetric ultrasound at the health centre on gestational age at birth.

Methods:

- in study endpoints where it is indicated that the study hypothesised that group ANC would increase mean GA at birth by 0·5±4·3 weeks, you should specify the statistic represented by "±4·3"; if it is not a range, then the ± designation should not be used.

- for the sample size calculation, you should indicate the cluster design-related parameters by which the sample size was inflated by to account for this design, with a justification for the choice.

- please organise the sub-sections in the methods to follow the sequence recommended in the CONSORT guidelines

- individual level bivariate analyses stratified by study group using Chi-square and Student’s t-test statistics are not appropriate for cluster-randomised trials because they do not adjust for clustering. In any case, statistical tests comparing groups in terms of baseline characteristics are not appropriate for randomised controlled trials.

- adjusting variables in a cRCT should be specified a priori and not based on post-hoc statistical considerations as described in the last paragraph of the methods.

Results:

- the CONSORT diagram is not appropriate for a cluster RCT; it should include the numbers of clusters at each stage and the mean or median number of women per cluster with the SD or IQR.

- the statistical tests comparing groups in table 3 are not appropriate in a cRCT

- avoid the ± designation all through the results and specify what the reported statistic is, whether range, SD or SE. For continuous outcomes, the mean and SD should be reported in the descriptive tables, and the mean and SE for inference (e.g. in Table 4, Table 5, Table 6, and the results of primary and secondary outcomes).

- see comment above about adjusting variables in randomised controlled trials.

Reviewer #3: This is an important paper summarising the results of a large cluster randomised controlled trial (RCT) assessing the effects of group antenatal care among women in Rwanda on child gestational length. The study is well-designed and thoroughly analysed, but I found it a bit difficult to follow. To do this research justice, I think that it should be published after the points below are addressed.

1. What’s the rationale for group vs individual antenatal care? The introduction includes a nice summary of the evidence available but does not mention the potential mechanism by which group care is hypothesised to improve gestational and perinatal outcomes.

2. While I appreciate that medical notes usually have many acronyms, people interested in reading this study may not necessarily know most of them, and reading a manuscript full of abbreviations can be quite time-consuming. Because PLOS One does not have a word count limit, I suggest keeping only a few universally known acronyms (e.g. RCT and WHO) and removing all the others from the text and figures.

3. Authors need to clarify the study aims from the beginning and to use descriptors consistently across the manuscript. For instance, the Methods section opens up with this sentence “To test the primary and secondary hypotheses [..]” although these hypotheses are described 5 pages later. I suggest adding a couple of sentences in the introduction to briefly mention the study hypotheses and related objectives, so that the reader has a clearer idea of the study aims. Also, are the primary hypothesis, primary outcome and primary analysis all linked somehow? If not, to avoid confusion I suggest using different adjectives, e.g. main analysis as opposed to ad-hoc analysis.

4. In the trial registration page (https://clinicaltrials.gov/ct2/show/NCT03154177) there is only one primary outcome, gestational age (“gestational length” in Table 4) and 8 secondary outcomes, whereas in the manuscript only 4 secondary outcomes are presented in tables 4 and 5. the following secondary outcomes are not presented in the main tables - apologies if I missed them somewhere else in the manuscript:

- Preterm 28-day and 42-day mortality rate

- Adherence to 6 week postnatal visit

- High-risk Women

- Newborns with neonatal morbidities

I suggest including all results in two tables, one with the findings of primary and secondary outcomes for the “group vs individual care” comparison and a very similar table for the “urine pregnancy testing vs ultrasound” comparison, while keeping the same outcome names using in the trial registration page.

5. This is a large-scale study and, understandably, the length of the Methods section reflects this. At the same time, I feel that some important details are missing. To preserve both readability and reproducibility, I suggest (i) moving most of the Methods section to a supplementary appendix, while keeping in the main text only the details that are essential to understand the Results and Discussion sections; and (ii) adding missing information to these supplementary Methods.

For example, the Sample Size section does not mention the following details

a) whether the trial was powered to detect an effect between group/individual care or between each of the 4 sturdy arms

b) what was the anticipated effect size relative to control arm(s), and from which previous studies was it obtained

c) the size of this inflation factor

d) the formula to calculate the final sample size from the inflation factor and the anticipated attrition constant, with a reference to the underlying methods paper

This trial has a large sample size in absolute terms, but if the anticipated effect size was very small it could still be underpowered, which could change the overall interpretation of the study. That is unlikely, but not impossible, and the reader currently does not have information to decide on this matter. While the most technical parts of this paragraph could be moved to the supplementary materials, an abridged version could stay in the main text, as it would be relevant to interpreting the study’s findings. An example PLOS One paper presenting the Methods in a more compact way and reporting the full protocol in the Appendix is available here: https://journals.plos.org/plosone/article?id=10.1371/journal.pone.0080561#s2

MINOR CONCERNS

This is a factorial cluster RCT, and the design could be mentioned in the title and the abstract.

Page 4. Ref #4 does not include the meta-analysis mentioned in the text. It looks like that would be reference #8 of this paper which includes the following sentence: “However, when the analysis was limited to the high quality studies (1 RCT and 1 observational study), African American women participating in group care had a significantly lower rate of preterm birth (2 studies (7, 8): pooled rates 8.0% vs. 11.1%, pooled RR 0.55; 95% CI 0.34–0.88).” consistently with what the authors report in the introduction of this paper. Please can authors replace reference #4 with https://pubmed.ncbi.nlm.nih.gov/27500348.

Page 7 Non-bipartite matching algorithm – to improve study reproducibility, please can you provide a specific reference (and/or the R package used for this, if applicable).

Page 10 Authors should briefly mention why a postnatal group visit was added, given that the primary purpose of the intervention was to improve gestational age.

Page 14. I agree with the choice to use generalized estimating equations model. To improve clarity, authors could explain in the method why the chose this method (I imagine to account for outcome correlation related to healthcare centres). An alternative could be using mixed effects models but this assumes a normal distribution of the data and this might not hold when using variables that are unlikely to be normally distributed, such as gestational age.

6. PLOS authors have the option to publish the peer review history of their article (what does this mean?). If published, this will include your full peer review and any attached files.

Reviewer #1: **Yes: **Christopher J. Doig

Reviewer #2: No

Reviewer #3: No

---

## [Author Response · Author response to Decision Letter 0]

4 Sep 2020

PONE-D-20-10387

Assessing the impact of group antenatal care on gestational length in Rwanda: a cluster-randomized trial

PLOS ONE

Response to Editor Comments

Journal Requirements:

Thank you for the templates. We have reviewed the style requirements and made corrections accordingly. 

2. You indicated that you had ethical approval for your study.

In your Methods section, please ensure you have also stated whether you obtained consent from parents or guardians of the minors included in the study or whether the research ethics committee or IRB specifically waived the need for their consent, or whether minors with children are legally allowed to consent for themselves.

 Regarding consent for minors, we have updated the Ethics statement in the Methods section to describe the IRB-approved waiver of parental consent as follows: Ethical approval was granted by the Rwanda National Ethics Committee (No.0034/RNE/2017) and University of California, San Francisco Institutional Review Board (16-21177). Data were collected on women aged 15 and older presenting for ANC at the 36 study health care centers who provided written informed consent between May 25, 2017 and December 31, 2018. This study was permitted by the Rwanda National Ethics Committee to waive parental consent for pregnant minors ages 15 and older. No pregnant adolescents younger than 15 years were documented to have been invited to participate in the study.

 Regarding the data availability statement, we will make the minimal data set available on datadryad.org at https://doi.org/10.7272/Q67W69F1

4. Please note that in order to use the direct billing option the corresponding author must be affiliated with the chosen institute. Please either amend your manuscript or remove this option (via Edit Submission).

 As a grantee of the Bill and Melinda Gates Foundation, it was our understanding that publishing fees would be paid directly by the Foundation. As such we have updated our record of this manuscript on the Chronos platform, the Foundation's portal to ensure open access publication and their payment of publication fees. If direct billing is not possible, please send an invoice to Elizabeth.butrick@ucsf.edu and we will submit it to the foundation.

5. We noted in your submission details that a portion of your manuscript may have been presented or published elsewhere:

'Table 1 was previously published in the Journal of Midwifery and Women's Health. Figure 1 was previously published in Gates Open Research. These publications are included as Related Manuscripts in this submission.'

Please clarify whether this publication was peer-reviewed and formally published.

If this work was previously peer-reviewed and published, in the cover letter please provide the reason that this work does not constitute dual publication and should be included in the current manuscript.

We appreciate the attention to detail; indeed, both Table 1 and Figure 1 have been previously published and peer reviewed. Because they do not include study data we thought it might be simpler to use the same information previously presented. Per your feedback, however, we have eliminated Table 1 from this manuscript, and inserted text with reference to the prior publication. Specifically, we state: During the second hour of the session, the provider and community health worker co-facilitated a discussion of health-related topics that aligned with the stage of pregnancy and the health issues of highest importance at that time. The full list of visit timing and topics at each visit are described elsewhere.13

Additionally, Figure 1 was previously published in our protocol paper. However, given some of the reviewers’ comments we have removed the figure as we believe it created confusion about the design. We have revised the text on page 6-7 to more accurately reflect the study design.

REVIEW COMMENTS TO THE AUTHOR

Reviewer #1: Thank you for the opportunity to read a well written paper on an important topic: addressing ANC and maternal/infant health in LMIC's/economically disadvantaged populations is very important. I do have some experience in healthcare in LMIC's in the African continent (but not Rwanda), cluster RCT's and GEE. I appreciate the complexity of the design, and the detailed explanations. I will focus more on technical issues related to design and analysis as this topic is not my clinical area of expertise.

1. This is a complicated design. I can not recall seeing an example of a cluster RCT with a factorial component. I remain unclear on aspects of your analysis. Usually a factorial design will report covariates for each factor, and an interaction estimate (between factors). The interaction effect is hopefully null, and then each factor effect can be reported independently. Was this done? Within a GEE, it is beyond my technical grasp how this would be setup given additional covariate for estimating random effects (intraclass correlations) as described in the manuscript. I think additional explanation is required, and perhaps supplemental analytical detail.

Thank you for highlighting this important design issue and giving us the opportunity to clarify the point. To clarify, the primary hypothesis of the study was not factorial, and the design was as a cluster randomized controlled trial. Conceptually, there is no reason to believe that ultrasound assessment would increase gestational age (the primary outcome) beyond the fact that early ultrasound assessment may rectify errors in LMP based GA. Further, there is no scientific evidence to further that line of argument. Additionally, ultrasound estimates of gestation were not included in the outcome variable. Thus, in line with our concept, and as stated in the protocol paper (https://gatesopenresearch.org/articles/3-1548) we have compared intervention (group antenatal care) vs. control (standard antenatal care). 

We acknowledge the ambiguity in Figure 1 and in the main text about the design of the study. In the revised version, we have included clarifying text and removed the figure. The main intervention was group ANC and the study was powered to measure the effect of group care only, as hypothesized a-priori in the protocol paper. The ultrasound component of the intervention was primarily to support quality of care and improved GA estimation. Thus, in accordance to the trial design, the analysis compared group ANC care with standard care, generically that is intervention vs. control, respectively. Please refer to the changes on page 6-7 of the main text.

We chose not to implement ultrasound in all sites for two reasons. First, with limited resources and thirty-six sites it was not practical. Second, we hoped to use those sites with ultrasound to develop a GA estimation algorithm to refine GA estimates in sites without ultrasound, which is addressed in another manuscript under development. Because we could not introduce ultrasound in all sites, we assigned each pair to a similar sized pair and randomized half the matched pairs to receive ultrasound, so that the introduction would not disrupt the group care matching and to produce a balance of sites with ultrasound across groups.

2. The pair matching is an additional complicated aspect of the design. As I read your methods, it was unclear to me if the 4 factors for matching were considered separately, or were considered as a composite. I think the 'range' of values which were considered a 'match' should be reported (e.g. 50%+/-5% was considered a match for proportion of first ANC visits, monthly delivery volume was 60 +/-10, et cetera). If matching variables were treated independently (and not as a composite) I'm not certain of the validity of reporting variables used in matching in Table 2 (unless to confirm no failure in matching) as variables used for matching must be within the specified parameters of the original matching unless there was a technical failure. I believe this would apply to the first 4 variables in Table 2.

We acknowledge the complicated nature of this step in the methods and have revised the Randomization and masking section to more clearly describe the process as follows:

To optimize group comparability, 36 health centers were pair-matched. First, the study statistician applied a non-bipartite matching algorithm in R, using the nbpMatching package (available from: https://cran.r-project.org/web/packages/nbpMatching/index.html), including data on monthly ANC volume, monthly delivery volume, proportion of ANC visits initiated before 16 weeks gestation, and availability of screening tests as a composite. Thirty sites were matched with the non-bipartite matching algorithm [24] with the remaining 6 presented as multiple options, which were then reviewed by the study team that finalized matching based on ANC volume, rural/urban designation and distance to the nearest hospital. After matching was complete, pairs were further matched to quadruples. Then one site in each pair was assigned to group ANC and the other to standard ANC using random selection in R software 19. One pair of each quadruple was similarly randomly selected for implementation of ultrasound and urine pregnancy testing.

As the reviewer points out, Table 2 serves to confirm there was no failure in matching. Because the matching was an algorithm that accounted for a composite of the parameters of interest, we have not provided ranges for individual parameters as they are balanced across the composite. We offer Table 2 as an easier to digest metric for understanding the success of the pairing than the ranges suggested above. If the editors think it more appropriate, we could move Table 2 to Supplemental material.

3. There is a risk of selection bias given the loss to follow-up. It is reassuring that the relative proportions reported in Figure 2 are somewhat similar. Was there any analysis performed on characteristics between mothers who completed the study compared to those where mother-baby pair data was available?

We acknowledge the concern for potential selection bias and have performed chi-square tests to analyze the difference between women included in the final sample and those excluded due to lack or return for 2 ANC or missing delivery data. While some of the socio-demographic data pointed to statistically significant differences between the populations, we believe these differences are a result of the large population size and high number of degrees of freedom, rather than important clinically significant differences as the magnitude of difference is generally quite small. We have included this table at the end of this document (Appendix A), should the reviewer desire to examine these data more closely. We are willing to include this data in Supplemental material if the editor feels it would be important.

4. Why was GL/GA not calculated based on the estimate from the first ANC rather than based at the time of delivery? Given entry into the study based on this estimate, and issues of blinding (although assessors were unblinded, it is an objective measure of GL/GA estimate if using the estimate from first ANC and date of delivery).

We appreciate this comment as this is an issue our team has discussed at length. In analyzing our data, we compared gestational length as estimated by maternity providers at delivery compared to gestational length calculated using the date of delivery and gestational age estimated at ANC1. Further we have analyzed these data in sites with ultrasound to evaluate the best “predictors” of gestational age and have a manuscript under development on this topic. We found that gestational length as estimated by the maternity provider at birth constituted the most complete dataset with the highest proportion of biologically plausible values.

The reasons for this decision are threefold. One, maternity providers have the most information available to them. They have the EDD as estimated from the ANC1 gestational length, and they are able to assess the newborn for birthweight and development at the time of delivery. Two, gestational length estimation at ANC1 is primarily based on LMP, which women may or may not recall accurately. Three, we found that because performance-based incentives in the health system in Rwanda incentivize providers for women enrolled in ANC prior to 16 weeks gestation, there is pressure on the providers, which is passed on to the patients for women to come in early. This pressure may well result in women reporting in lower GAs than is actually accurate at the time of initiation of ANC. The misclassification of gestational age toward lower gestational ages at ANC1 would result in higher preterm birth rates.

Minor comment

5. In the results section, I think more technically accurate language would be that participant characteristics were statistically significant, but were not considered to be practically or clinically significant (I'm not sure of a technical meaning for insubstantial). I'm not a fan of statistical testing of demographic features between treatment arms, and your study might be a good example of permitting the reader to draw their own conclusion as to if any of these features are practically significant (rather than being drawn to a p value that perhaps better reflects size of the study).

Thank you for the recommendation. We agree and have thus removed the p-values from Table 3 (now Table 2). We have revised the text on page 17 to read: While various differences between the study groups' participant characteristics were statistically significant, we do not consider these differences to be practically or clinically significant (Table 2).

Reviewer #2: Abstract:

6. since this was a cluster randomised trial, the reported findings should also have the number of clusters included in the analysis. You should report this alongside the numbers of women in each arm, e.g. "4091 women in X clusters and 4752 women in Y clusters at control and intervention facilities, respectively, were included..."

Thank you. We have revised the first sentence of “Abstract/Findings” to read: 4091 women in 18 clusters and 4752 women in 18 clusters at control and intervention facilities had outcomes analyzed. 

7. you need to specify the statistics represented by the "±1·6" and "±1·5" - is this the range of values? If so say so; if it is another statistic other than the range, please remove the ± designation.

Thank you for pointing this out. The statistic represented is the standard deviation. We have revised the sentence to read: Gestational length was equivalent in the intervention and control groups (39.3 weeks (SD 1.6) and 39.3 weeks (SD 1.5)).

Objectives:

8. the objectives make no mention of the ultrasound component of the intervention; indeed having read it, the description of the trial seems incomplete to me - it is a factorial cluster randomised trial testing the effect of group ANC with or without obstetric ultrasound at the health centre on gestational age at birth.

Please see Comment #1 for a detailed response to this issue. The ultrasound and community-based pregnancy testing, which were only expected to impact secondary outcomes, are mentioned in the last sentence of the Objectives section on page 5, and described in the Study Interventions section on page 10.

Methods:

9. in study endpoints where it is indicated that the study hypothesised that group ANC would increase mean GA at birth by 0·5±4·3 weeks, you should specify the statistic represented by "±4·3"; if it is not a range, then the ± designation should not be used.

Thank you for pointing this out. The statistic represented is the standard deviation. We have removed the ± designation throughout. We have revised the sentence, now in the Outcomes section on page 11 to read: …would increase mean gestational length by .5 weeks (with a standard deviation no larger than 4.3 weeks) compared to…

10. for the sample size calculation, you should indicate the cluster design-related parameters by which the sample size was inflated by to account for this design, with a justification for the choice.

At the design stage of this trial, we assumed an ICC of 0.01, based on the literature for conventional antenatal care versus group antenatal care. We assumed a loss-to-follow-up rate of 30-50%, working on the assumptions that 10-20% of all pregnancies end in miscarriage, and that 20-30% of women might deliver at another location. Given these assumptions, for 36 facilities and 222 deliveries per facility, the design effect that this study would have accounted for was 3.21. The study was powered to detect a 0.5 week (with a standard deviation of 4.3 weeks) difference in gestational length. For enhanced clarity, we have revised the sample size section of the manuscript, on page 13, copied below:

Sample Size:

At the design stage of this trial, we assumed an ICC of 0.01, and a possible effect size of half a week in gestational age based on a Cochrane review of conventional versus group antenatal care.[22] We assumed a loss-to-follow-up rate of 30%, based on the assumptions that 10-15% of all pregnancies end in miscarriage and 15-20% of women might deliver at another location. We calculated that for a two-tailed test, α=0·05, 1-β=80%, and a balanced 1:1 ratio of intervention (group antenatal care) to control (standard antenatal care) study participants, a minimum sample of n=1,163 eligible women per study group (intervention and control) was required. The sample size was increased to account for a cluster design effect of 3.21 (cluster randomization to study group by health center rather than by individual) and to account for a loss to follow-up rate of up to 50%, for a total required sample size per study group of 3,640.[23] As descriptive outcomes, the secondary outcomes were assessed in the available sample without hypothesis testing or multiple comparisons adjustment.

The ICC from actual data was much lower (0.00063) and with 246 deliveries per facility the actual design effect was 1.15. However, the observed difference was much lower (e.g., nearly null), 0.07 weeks, than the difference that this study was powered to detect. 

11. please organise the sub-sections in the methods to follow the sequence recommended in the CONSORT guidelines

We have reordered the sub-sections in the methods section to more seamlessly follow the sequence recommended in the CONSORT guidelines. Thank you for the recommendation.

12. individual level bivariate analyses stratified by study group using Chi-square and Student’s t-test statistics are not appropriate for cluster-randomised trials because they do not adjust for clustering. In any case, statistical tests comparing groups in terms of baseline characteristics are not appropriate for randomised controlled trials.

Thank you for this comment. We agree that the lack of adjustment for clustering may be misleading. Another reviewer made a similar comment and thus we have opted to remove the p-values from Table 2. Please see Comment #5 for additional detail.

13. adjusting variables in a cRCT should be specified a priori and not based on post-hoc statistical considerations as described in the last paragraph of the methods.

We have not adjusted for any confounder in the final models in both the original and the revised version (Table 3). The two factors that we did adjust for relate to design aspects. The CRCT was pair-matched, hence the pairing had to be accounted for in the models. Further, the primary outcome was gestational length, which was measured for each birth, which in turn were clustered within facilities. Hence, we accounted for that clustering structure of the data in the statistical models. In other words, we only adjusted for “pairing” and “clustering.”

The adjustment that we mentioned in the last paragraph of the methods on page 18 of the original version, was part of the sensitivity analysis to demonstrate robustness of the results. We have modified the text to clarify this point, on page 15 of the revised manuscript:

“We developed a Directed Acyclic Graph to identify potential confounders and mediators for which data were available. In descriptive analyses, we investigated the factors on which the intervention and control arms were statistically significantly different. The final model was adjusted for pairing and clustering only. In sensitivity analyses, we additionally adjusted for potential confounders or mediators to demonstrate that the results were unaffected by residual confounding that may have resulted due to unbalanced study arms.”

In addition, we would like to note that given the null results, it is unlikely that any single or group of suppressor variables would have explained the observed results.

Results:

14. the CONSORT diagram is not appropriate for a cluster RCT; it should include the numbers of clusters at each stage and the mean or median number of women per cluster with the SD or IQR.

Thank you for drawing our attention to this important omission. The CONSORT diagram has been revised accordingly.

15. the statistical tests comparing groups in table 3 are not appropriate in a cRCT

Another reviewer made a similar comment and thus we have removed the p-values from Table 2. Please see comment #5 for additional detail.

16. avoid the ± designation all through the results and specify what the reported statistic is, whether range, SD or SE. For continuous outcomes, the mean and SD should be reported in the descriptive tables, and the mean and SE for inference (e.g. in Table 4, Table 5, Table 6, and the results of primary and secondary outcomes).

Thank you for this comment. For all descriptive estimates such as means, we have removed the ± designation throughout and reported SDs. For inferential estimates (i.e. model based results) we reported the confidence intervals in Tables 3-4.

17. see comment above about adjusting variables in randomised controlled trials.

Please see response above to comment # 13.

Reviewer #3: This is an important paper summarising the results of a large cluster randomised controlled trial (RCT) assessing the effects of group antenatal care among women in Rwanda on child gestational length. The study is well-designed and thoroughly analysed, but I found it a bit difficult to follow. To do this research justice, I think that it should be published after the points below are addressed.

18. What’s the rationale for group vs individual antenatal care? The introduction includes a nice summary of the evidence available but does not mention the potential mechanism by which group care is hypothesised to improve gestational and perinatal outcomes.

Thank you for this comment. We have added the rationale in the introduction at the end of the first paragraph, appending the following sentence:

Group ANC is hypothesized to positively impact preterm birth rates and other outcomes among women at elevated psychosocial risk due to 3 main features of the model: 1) greater social support between women who are linked via the group; 2) more total ANC-associated time spent in educational activities in facilitated group discussions; and 3) attention to key elements of person-centered care, including respect and safety, empowerment, and participation.5–7 These elements create a more positive pregnancy care experience which may encourage ANC attendance and thus create additional opportunity for risk assessment by providers.

19. While I appreciate that medical notes usually have many acronyms, people interested in reading this study may not necessarily know most of them, and reading a manuscript full of abbreviations can be quite time-consuming. Because PLOS One does not have a word count limit, I suggest keeping only a few universally known acronyms (e.g. RCT and WHO) and removing all the others from the text and figures.

We have spelled out the following acronyms in the text ANC, PNC, PTBi, LMIC, MOH, UPT, PTB, LBW, SGA, REDCap, GA, and GEE. We retained the acronyms: RCT, WHO, SPSS, STATA, and HIV, and “R” software

We also changed “gestational age at birth” to “gestational length” throughout text to be consistent with article title.

20. Authors need to clarify the study aims from the beginning and to use descriptors consistently across the manuscript. For instance, the Methods section opens up with this sentence “To test the primary and secondary hypotheses [..]” although these hypotheses are described 5 pages later. I suggest adding a couple of sentences in the introduction to briefly mention the study hypotheses and related objectives, so that the reader has a clearer idea of the study aims. Also, are the primary hypothesis, primary outcome and primary analysis all linked somehow? If not, to avoid confusion I suggest using different adjectives, e.g. main analysis as opposed to ad-hoc analysis.

Thank you. We have made a number of revisions to improve clarity and consistency related to our study aims. 

As per the study's a priori hypotheses, this paper reports on primary and secondary outcomes of our CRCT only. The primary analysis presents gestational length, the study primary outcome, on which the study was powered. The intervention tested is the effect of group antenatal care on gestational length at birth. We also analyzed secondary outcomes, as outlined in our protocol paper including the effect of group antenatal care on rates of preterm birth, mortality among preterm neonates, attendance at 4 ANC visits, initiation of ANC before 14 weeks, attendance at 6-week postnatal care visits, identification of women at high risk during pregnancy, and cesarean section. We are unable to report on newborn morbidities as planned due to data quality issues. 

In addition to the intervention of interest, group antenatal care, both ultrasound examination by health center providers and community-based urine pregnancy testing were implemented in half of the sites in each study group. Additional secondary analyses were conducted to see if group antenatal care, ultrasound examination and/or community-based urine pregnancy testing were associated with the secondary outcomes of attendance at 4 ANC visits and initiation of ANC before 14 weeks.

We have made modifications to the paper to help improve clarity including:

• In the “objectives” subsection of the Introduction: The sentence “Intrigued by lower rates of preterm birth among high-risk American women who participated in group antenatal care, the Preterm Birth Initiative-Rwanda aimed to test the primary [added ‘primary’] hypothesis that Rwandan women receiving antenatal care at health centers that offer group antenatal care would experience increased gestational length compared to women receiving antenatal care at health centers that provide the standard, individual model of care. Rwandan stakeholders preferred to offer all women at each health center the same model of care.” 

• Added to this paragraph, on page 5-6: We also explored whether group antenatal care affected the secondary outcomes of preterm birth, mortality among preterm neonates, attendance at 4 antenatal care visits, attendance at the first antenatal care visit before 14 completed weeks gestation, attendance at a six-week postnatal care visit at health facilities, identification of women as at high risk at any antenatal care visit, and caesarean section rates among enrolled women. We intended to examine newborn morbidities but our data sources were not adequate to do so. Further, at half of the health centers included in this study we implemented community-based urine pregnancy test by community health workers and basic obstetric ultrasound by nurses and midwives and conducted secondary analyses to see if these interventions affected the secondary outcomes of attendance at 4 antenatal care visits and initiation of antenatal care before 14 completed weeks.

21. In the trial registration page (https://clinicaltrials.gov/ct2/show/NCT03154177) there is only one primary outcome, gestational age (“gestational length” in Table 4) and 8 secondary outcomes, whereas in the manuscript only 4 secondary outcomes are presented in tables 4 and 5. the following secondary outcomes are not presented in the main tables - apologies if I missed them somewhere else in the manuscript:

- Preterm 28-day and 42-day mortality rate

- Adherence to 6 week postnatal visit

- High-risk Women

- Newborns with neonatal morbidities

I suggest including all results in two tables, one with the findings of primary and secondary outcomes for the “group vs individual care” comparison and a very similar table for the “urine pregnancy testing vs ultrasound” comparison, while keeping the same outcome names using in the trial registration page.

Thank you for your careful review of both our paper and our protocol. We have revised the tables according to the reviewer’s suggestions, as we agree it is important to report on the primary and all the listed secondary outcomes. 

The data for the outcomes preterm mortality and morbidity is too scanty to reliably make any inferences, and we have noted as much in the revised manuscript. 

However, we have reported figures on newborn mortality, though the majority of the available data was at birth, rather than at 28 and 42 days as planned, due to insufficient follow-up data. We were unable to systematically collect newborn morbidity data and thus have left it out of the table, and inserted an explanation on page 12 of the omission and addressed it in the discussion. 

These data limitations were due, at least in part, to the fact that one of the expected data sources was the national SMS reporting of newborn morbidities and mortalities by community health workers. Unlike all of our other data sources, where study data collectors could access paper copies of facility registers and patient files without impeding the work of health center staff, retrieving SMS data required significant time and cooperation from facility-based data managers. Our data collection team was unable to effectively gain the systematic cooperation of these data managers, and thus in our data system not a single newborn morbidity report appeared. Further, reports from our field staff reveal that the Rapid SMS system was often down in rural areas, limiting their ability to access this data. Rather than report this patently deficient data point, we opted to omit it from the table, and have instead acknowledged the limitation in the text on page 26.

As suggested, we have included all the secondary outcomes in the revised main outcomes table. The outcomes table for the analysis by ultrasound and pregnancy testing versus no ultrasound or pregnancy testing contains only the secondary outcomes of gestational age at entry to care and proportion of women attending at least 4 antenatal visits, as these are the only outcomes these were expected to be associated with.

22. This is a large-scale study and, understandably, the length of the Methods section reflects this. At the same time, I feel that some important details are missing. To preserve both readability and reproducibility, I suggest (i) moving most of the Methods section to a supplementary appendix, while keeping in the main text only the details that are essential to understand the Results and Discussion sections; and (ii) adding missing information to these supplementary Methods.

For example, the Sample Size section does not mention the following details

a) whether the trial was powered to detect an effect between group/individual care or between each of the 4 sturdy arms 

b) what was the anticipated effect size relative to control arm(s), and from which previous studies was it obtained 

c) the size of this inflation factor 

d) the formula to calculate the final sample size from the inflation factor and the anticipated attrition constant, with a reference to the underlying methods paper 

This trial has a large sample size in absolute terms, but if the anticipated effect size was very small it could still be underpowered, which could change the overall interpretation of the study. That is unlikely, but not impossible, and the reader currently does not have information to decide on this matter. While the most technical parts of this paragraph could be moved to the supplementary materials, an abridged version could stay in the main text, as it would be relevant to interpreting the study’s findings. An example PLOS One paper presenting the Methods in a more compact way and reporting the full protocol in the Appendix is available here: https://journals.plos.org/plosone/article?id=10.1371/journal.pone.0080561#s2

Thank you. These points have been addressed in Comment #10 in response to another reviewer.

Further, we appreciate the comment on the power. The design effect for the ICC of the study sample (0.00063 and 8843 respectively) for 36 clusters (246 women per cluster) is 1.15435. We originally assumed an ICC of 0.001, so, although we had a smaller ICC, we also had a much smaller effect than the 0.5 week effect we designed for. When designing the trial we chose the 0.5 week effect size as the smallest clinically relevant effect we thought might be attainable.

MINOR CONCERNS

23.This is a factorial cluster RCT, and the design could be mentioned in the title and the abstract.

Please see comment #1. We appreciate this perspective, but have retained the title as the study was designed as a CRCT.

24. Page 4. Ref #4 does not include the meta-analysis mentioned in the text. It looks like that would be reference #8 of this paper which includes the following sentence: “However, when the analysis was limited to the high quality studies (1 RCT and 1 observational study), African American women participating in group care had a significantly lower rate of preterm birth (2 studies (7, 8): pooled rates 8.0% vs. 11.1%, pooled RR 0.55; 95% CI 0.34–0.88).” consistently with what the authors report in the introduction of this paper. Please can authors replace reference #4 with https://pubmed.ncbi.nlm.nih.gov/27500348.

Thank you for your attention to detail. This has been corrected in the revised manuscript.

25. Page 7 Non-bipartite matching algorithm – to improve study reproducibility, please can you provide a specific reference (and/or the R package used for this, if applicable).

Thank you. Please see Comment #2 for additional discussion on this topic. Our revised text there includes a reference to Lu et al 2011, (Optimal Nonbipartite, Matching and Its Statistical Applications, The American Statistician, 65:1, 21-30). We have also included a link to the nbpMatching package:https://cran.r-project.org/web/packages/nbpMatching/index.html

26 Page 10 Authors should briefly mention why a postnatal group visit was added, given that the primary purpose of the intervention was to improve gestational age.

Thank you for this insightful comment. We have revised a sentence in second paragraph of Group Antenatal Care and Postnatal Care subsection to include this explanation:

The Technical Working Group hoped that the social support fostered among women in the same antenatal group would continue into the postnatal period and motivate women to seek care; for this reason, a postnatal group visit was included in the model even though a postnatal visit was not expected to impact the primary outcome (gestational length).

27. Page 14. I agree with the choice to use generalized estimating equations model. To improve clarity, authors could explain in the method why the chose this method (I imagine to account for outcome correlation related to healthcare centres). An alternative could be using mixed effects models but this assumes a normal distribution of the data and this might not hold when using variables that are unlikely to be normally distributed, such as gestational age.

We appreciate the concurrence of the reviewer with our analytical approach. The robust GEE and the mixed effect models are the two suitable options to analyze this data. We chose the former because we wanted to generate robust estimates that controlled for design effect. We chose the GEE to generate one overall intervention effect across all pairs of facilities. In addition, to the point the reviewer highlighted, we would like to state that our interest was not to generate intervention effect separately for each pair of facilities, or how this effect might have differed between pairs, in which case a mixed effect model with a random intercept/slope for each pair would have been relevant.

---

## [Decision Letter · Decision Letter 1]

3 Nov 2020

PONE-D-20-10387R1

Assessing the impact of group antenatal care on gestational length in Rwanda: a cluster-randomized trial

PLOS ONE

Dear Dr. Lundeen,

Thank you for submitting your manuscript to PLOS ONE. After careful consideration, we feel that it has merit but does not fully meet PLOS ONE’s publication criteria as it currently stands. Therefore, we invite you to submit a revised version of the manuscript that addresses the points raised during the review process.

We look forward to receiving your revised manuscript.

Kind regards,

Seth Adu-Afarwuah

Academic Editor

PLOS ONE

Reviewers' comments:

Reviewer's Responses to Questions

**Comments to the Author**

1. If the authors have adequately addressed your comments raised in a previous round of review and you feel that this manuscript is now acceptable for publication, you may indicate that here to bypass the “Comments to the Author” section, enter your conflict of interest statement in the “Confidential to Editor” section, and submit your "Accept" recommendation.

Reviewer #1: (No Response)

Reviewer #2: All comments have been addressed

2. Is the manuscript technically sound, and do the data support the conclusions?

Reviewer #1: Yes

Reviewer #2: Yes

3. Has the statistical analysis been performed appropriately and rigorously? 

Reviewer #1: Yes

Reviewer #2: Yes

4. Have the authors made all data underlying the findings in their manuscript fully available?

Reviewer #1: Yes

Reviewer #2: Yes

5. Is the manuscript presented in an intelligible fashion and written in standard English?

Reviewer #1: Yes

Reviewer #2: Yes

6. Review Comments to the Author

Reviewer #1: Thank you for your thoughtful comments. I've had the opportunity to review responses to my comments, and the comments of other reviewers. I'd like to congratulate you on conducting an important study with an elegant design.

Reviewer #2: Abstract: correct the typo in the first line of 'Findings'; I suggest it reads: "A total of 4091 women in 18 control clusters and 4752 women in 18 intervention clusters were included in the analysis..."

Background line 4: "an individually randomized" instead of "individual randomized"

Objectives: a lot of the explanations now included in the objectives, although very useful, do not belong there. For example, the following sentence should be included in the justification for the cluster-randomised design, such as in the second sentence of the 'trial design' section (words in brackets are my own suggestion): "(We opted for a clustered design because) Rwandan stakeholders preferred to offer all women at each health center the same model of care." The following sentence should also be moved to an appropriate place in the methods: "Further, at half of the health centers included in this study we implemented community-based urine pregnancy test by community health workers and basic obstetric ultrasound by nurses and midwives and conducted secondary analyses to see if these interventions affected the secondary outcomes of attendance at four antenatal care visits and initiation of antenatal care before 14 completed weeks."

Methods: description of the sample size calculation is still inadequate. This was a cluster-randomised trial; the description should include at least two of the following: the number of clusters per arm; the number of observations per cluster; the number of individuals per arm. At the moment, only the number of individuals per arm is included, giving the reader no idea of the number of clusters per arm or observations per cluster. Please look at the reporting in other examples of studies of the same design e.g. 10.1371/journal.pmed.1001018.

According to your response to peer review, you are no longer doing this, therefore this should be removed from your 'statistical analysis' section: "We conducted individual level bivariate analyses stratified by study group using Chi-square and Student’s t-test statistics for categorical and continuous data, respectively, to assess study group comparability. Similar unadjusted bivariate analyses were conducted for the primary and secondary outcomes." - please go through the whole manuscript carefully to ensure that the changes you have made are reflected in the latest version and are consistent all through.

Results: for the description of the sample in the first paragraph of the results, please report the mean and range or median and IQR per cluster, not mean and SD - the SD is not very meaningful in this context (fine to report SD elsewhere). Your CONSORT diagram is still incomplete - it should include the number of clusters with mean+range or median+IQR of women per cluster all through; see the CONSORT diagram in 10.1371/journal.pmed.1001018 for example.

When you report the continuous outcomes, you should use standard errors (SE) instead of standard deviations (SD); SDs are for description, SEs are for inference (and you are making inference on the outcomes).

7. PLOS authors have the option to publish the peer review history of their article (what does this mean?). If published, this will include your full peer review and any attached files.

Reviewer #1: **Yes: **Christopher James Doig, Professor, Departments of Critical Care Medicine, and Community Health Sciences, University of Calgary

Reviewer #2: No

---

## [Author Response · Author response to Decision Letter 1]

25 Nov 2020

PONE-D-20-10387R1

Assessing the impact of group antenatal care on gestational length in Rwanda: a cluster-randomized trial

PLOS ONE

RESPONSE TO REVIEWERS

Reviewer #1: Thank you for your thoughtful comments. I've had the opportunity to review responses to my comments, and the comments of other reviewers. I'd like to congratulate you on conducting an important study with an elegant design.

Reviewer #2: Abstract: correct the typo in the first line of 'Findings'; I suggest it reads: "A total of 4091 women in 18 control clusters and 4752 women in 18 intervention clusters were included in the analysis..."

Thank you for your careful attention to detail. We have corrected the typo and appreciate the suggestion.

Background line 4: "an individually randomized" instead of "individual randomized"

Corrected.

Objectives: a lot of the explanations now included in the objectives, although very useful, do not belong there. For example, the following sentence should be included in the justification for the cluster-randomised design, such as in the second sentence of the 'trial design' section (words in brackets are my own suggestion): "(We opted for a clustered design because) Rwandan stakeholders preferred to offer all women at each health center the same model of care." The following sentence should also be moved to an appropriate place in the methods: "Further, at half of the health centers included in this study we implemented community-based urine pregnancy test by community health workers and basic obstetric ultrasound by nurses and midwives and conducted secondary analyses to see if these interventions affected the secondary outcomes of attendance at four antenatal care visits and initiation of antenatal care before 14 completed weeks." 

Thank you for the suggestions. In rereading we understand that some of the additions we put in in response to previous reviewer comments may have overcompensated. We further appreciate the specificity of this reviewer’s suggestions and have made the following changes:

As suggested, we moved the explanation of Rwandan stakeholders’ preference for uniform care at any health center to the trial design justification, on page 6. We have also edited this section to address only the objectives and moved the information about ultrasound and urine pregnancy section to the methods, describing in trial design the distribution of these interventions in a balanced manner on page 6-7.

Methods: description of the sample size calculation is still inadequate. This was a cluster-randomised trial; the description should include at least two of the following: the number of clusters per arm; the number of observations per cluster; the number of individuals per arm. At the moment, only the number of individuals per arm is included, giving the reader no idea of the number of clusters per arm or observations per cluster. Please look at the reporting in other examples of studies of the same design e.g. 10.1371/journal.pmed.1001018. 

We have revised the sample size section on page 13 to reflect the underlying assumption of 36 health centers total, or 18 per study group, and an average of 202 women per cluster.

According to your response to peer review, you are no longer doing this, therefore this should be removed from your 'statistical analysis' section: "We conducted individual level bivariate analyses stratified by study group using Chi-square and Student’s t-test statistics for categorical and continuous data, respectively, to assess study group comparability. Similar unadjusted bivariate analyses were conducted for the primary and secondary outcomes." - please go through the whole manuscript carefully to ensure that the changes you have made are reflected in the latest version and are consistent all through. 

Thank you. We apologize for the confusion. We did conduct bivariate analysis for comparability, but removed the Chi-square and t-tests, as well as all unadjusted results for outcomes, and have now thoroughly reviewed the manuscript for these inconsistencies and removed references to results that are not presented in the paper.

Results: for the description of the sample in the first paragraph of the results, please report the mean and range or median and IQR per cluster, not mean and SD - the SD is not very meaningful in this context (fine to report SD elsewhere). Your CONSORT diagram is still incomplete - it should include the number of clusters with mean+range or median+IQR of women per cluster all through; see the CONSORT diagram in 10.1371/journal.pmed.1001018 for example. 

Thank you. We appreciate the clarifications from this reviewer and apologize for the confusion. We have replaced SDs with ranges in the first paragraph of the results section, and updated the CONSORT throughout to include cluster size and ranges at each level.

When you report the continuous outcomes, you should use standard errors (SE) instead of standard deviations (SD); SDs are for description, SEs are for inference (and you are making inference on the outcomes).

Thank you for the clarification. We have updated the continuous outcomes in Tables 3, S1, and S3 to provide SEs rather than SDs for continuous outcomes as well as more clearly labelling the column headers for the reader.

We have similarly cleaned up the column headers in Tables 1 and 4 for clarity.

---

## [Decision Letter · Decision Letter 2]

20 Jan 2021

Assessing the impact of group antenatal care on gestational length in Rwanda: a cluster-randomized trial

PONE-D-20-10387R2

Dear Dr. Lundeen,

We’re pleased to inform you that your manuscript has been judged scientifically suitable for publication and will be formally accepted for publication once it meets all outstanding technical requirements.

Kind regards,

Seth Adu-Afarwuah

Academic Editor

PLOS ONE

Additional Editor Comments (optional):

Reviewers' comments:

Reviewer's Responses to Questions

**Comments to the Author**

1. If the authors have adequately addressed your comments raised in a previous round of review and you feel that this manuscript is now acceptable for publication, you may indicate that here to bypass the “Comments to the Author” section, enter your conflict of interest statement in the “Confidential to Editor” section, and submit your "Accept" recommendation.

Reviewer #2: All comments have been addressed

2. Is the manuscript technically sound, and do the data support the conclusions?

Reviewer #2: (No Response)

3. Has the statistical analysis been performed appropriately and rigorously? 

Reviewer #2: (No Response)

4. Have the authors made all data underlying the findings in their manuscript fully available?

Reviewer #2: (No Response)

5. Is the manuscript presented in an intelligible fashion and written in standard English?

Reviewer #2: (No Response)

6. Review Comments to the Author

Reviewer #2: (No Response)

7. PLOS authors have the option to publish the peer review history of their article (what does this mean?). If published, this will include your full peer review and any attached files.

Reviewer #2: No

---

## [Editor Report · Acceptance letter]

22 Jan 2021

PONE-D-20-10387R2 

Assessing the impact of group antenatal care on gestational length in Rwanda: a cluster-randomized trial 

Dear Dr. Lundeen:

I'm pleased to inform you that your manuscript has been deemed suitable for publication in PLOS ONE. Congratulations! Your manuscript is now with our production department. 

Kind regards, 

on behalf of

Dr. Seth Adu-Afarwuah 

Academic Editor

PLOS ONE